# Ni-Cr Alloys Assessment for Dental Implants Suitability

Dragos Cristian Achitei [1], Mădălina Simona Baltatu [1,*], Petrică Vizureanu [1,2], Andrei Victor Sandu [1,3], Marcelin Benchea [4,*] and Bogdan Istrate [4]

1 Department of Technologies and Equipments for Materials Processing, Faculty of Materials Science and Engineering, Gheorghe Asachi Technical University of Iaşi, Blvd. Mangeron, No. 51, 700050 Iasi, Romania
2 Technical Sciences Academy of Romania, Dacia Blvd. 26, 030167 Bucharest, Romania
3 National Institute for Research and Development in Environmental Protection INCDPM, Splaiul Independentei 294, 060031 Bucharest, Romania
4 Faculty of Mechanical Engineering, Gheorghe Asachi Technical University of Iaşi, Blvd. Mangeron, No. 51, 700050 Iasi, Romania
* Correspondence: cercel.msimona@yahoo.com (M.S.B.); marcelin.benchea@tuiasi.ro (M.B.)

**Abstract:** The performance of the field and the condition of success in oral implantology today require the review and reevaluation of the means that contribute essentially to ensuring the stability and durability of the implant, starting from the nature of the biomaterial and continuing with the characteristics of the optimally designed biosurface. This paper proposes a comparative analysis of three commercial alloys, VeraBond, Kera N, and VeraSoft, compared to a modified dental alloy, with the aim of improving some mechanical properties. They have been studied structurally and mechanically. The microstructural structure shows that the alloys crystallize in the face-centered cube system, and the cast alloy has a dendritic structure with large grains. XRD diffractograms highlight that alloys exhibit three compounds $Cr_{156.00}Al_{596.00}$: 9013031, $Ni_{4.00}$, and $Cr_{30.00}Al_{48.00}$. The hardness measurements showed values between 203 HV and 430 HV. As the percentage of silicon increases, the hardness decreases. The modulus of elasticity obtained by the indentation method for the dental alloys was in the range of 46–153 GPa. The results showed that the hardness and elastic modulus of the new alloy was significantly minimized compared to the classical alloys used.

**Keywords:** Ni-Cr; properties; cast samples; dental alloys





## 1. Introduction

The dental implant, with all its forms, can be defined as a body foreign to the body (alloplastic material), which is surgically inserted at the level of the jawbone, with the main purpose of supporting the prosthetic elements used in the treatment of edentulousness. Unlike other types of medical implants, the dental implant is considered "open" due to the communication between it and the oral cavity, which is a septic environment with many potential factors of aggression [1,2].

Current research regarding the modification and control of the biomaterial–tissue interaction to improve the osseointegration process of the implant is oriented towards capitalizing on the progress made in regenerative medicine [3–5], using tissue engineering techniques and developing the field of biomimetic materials (so-called smart biomaterials) [6–9].

The applications of biomaterials in the field of medicine are primarily due to the requirements imposed by medical practice [10,11] but also by the continuous evolution of science. A permanent correlation of research in the fields of chemistry, biology, engineering, and medicine leads the science of biomaterials to obtain new materials that can solve the multiple existing medical problems to the current requirements [12].

The physical, chemical and mechanical properties of dental materials are of particular importance to support the desire to forecast and design new devices and material assemblies. Thus, the review of these properties has the aim of creating a specific framework for

starting the complete study of dental materials; without the knowledge of these data, new designs will be stuck or simply deviate from the expected results [13,14].

It has been proven that the general properties and the surface properties of the metals that have been used for implants directly influence and, in some cases, even control the dynamics at the tissue interface from the time of initial placement on the live to the final removal. It is admitted that compatibility is a two-way process between the biomaterials incorporated in the device and the implant host environment [15,16].

Remarkable advances have been made in dentistry. In the last 25 years, the rapid progress of discoveries regarding biomaterials and implants has only been possible through close collaboration between doctors, biologists, and engineers. Both the improvement in the types of implants and the discovery of new materials that are best tolerated by the body were pursued [17–19].

Ni-Cr alloys have appeared as an alternative to Co-Cr alloys that present a low ductility, increase shrinkage upon solidification, and have a tendency to oxidize. In the industry, these alloys are known as NIMONIC and are used in the construction of reaction engines [20–22].

Ni-Cr alloys contain up to 70% nickel. Several types of alloys containing Ni are used in dentistry:

- Ni-Cr-Fe alloys (wipla type, classic) with a percentage of 48–66%Ni;
- Ni-Cr alloys also contain small percentages of Mo, Al, Mn, Be, Cu, Co, Ga, and Fe to improve certain properties of the alloy.
- Co-Cr-Ni alloys are used in skeletal prosthesis technology [23–25].

Biocompatibility is a very important aspect of alloys that have medical applications. The release of metal ions from these alloys is determined by the percentage of chromium and must be above 20% for proper passivation of the alloy. Chromium, as an element in the composition of an alloy, changes the melting temperature and increases the mechanical properties, and by forming oxides, it increases the corrosion resistance and the bond with ceramics.

Nickel is an essential element for the human body, containing about 10 mg of Ni. The daily intake is recommended to be 100 μg per day, mainly through food. Nickel compounds are generally very water-soluble, and therefore, in the case of corrosion, they are quickly entrained by saliva in the intestinal tract. For this reason, the contact time with the Langerhans cells present in the oral mucosa and acting as receptors is very short. Due to its nature and chemical structure, the oral mucosa allows for faster diffusion of Ni ions than the skin, and the risk of sensitization in the oral cavity is extremely low. There is currently no description of Ni accumulations in the body. Despite the relatively frequent allergies to skin contact with Ni, it has been clinically observed that the use of Ni-Cr alloys in the oral cavity does not systematically cause allergic reactions [26–28].

If we were to consider only the initial corrosion in the oral environment of the stable Ni-Cr alloy (e.g., Wiron88/Bego), in order to reach the recommended daily level of Ni absorption of 100 μg, it would result in 25 cm$^2$ of this alloy. It should also be taken into account that the initial corrosion phenomenon, namely the release of ions, decreases rapidly in the first hours and days, and after a few days, much fewer ions are released. We can conclude that a possible sensitization occurs only in very sensitive people. Given the small amounts released and the short biological half-life of Ni, a systemic toxic attack must be excluded. However, we cannot exclude a local toxic action, which can also occur for other metals. Such a phenomenon can occur in the case of incorrect finishing by the technician of the prosthetic piece.

The use of metallic biomaterials in the human body is dependent on certain specific characteristics of the material, as well as on the specific function it has to perform.

Subsequently, through improvements to the materials and design, the success rate increased. The concurrent development of asepsis and antisepsis protocols and anesthetic techniques also contribute greatly to this good outcome. After 1980, implantology aroused more and more interest in doctors from many corners of the world, and various congresses have been organized on this topic (important to mention the one in Toronto in 1982 where

Brånemark published the results of his research for more than 15 years in the field of osseointegration). From this moment, implantology begins to make unexpected progress. The number of dental implants applied worldwide is still increasing year by year [29,30].

We carried out a preliminary study in order to follow the opportunity of introducing a new type of dental implant. The new alloy was designed so that, by eliminating the harmful effects present in the dental alloys frequently used today in oral implantology, we could control the intrinsic biocompatibility of the material: a character with a defining influence on the osseointegration process of the implant by adding silicon. We choose the solution of alloying with an element such as silicon to improve the mechanical properties [31].

The aim of this work was to change the percentage of the metallic silicon of the VeraSoft-type dental alloy from the percentage of 1.60% to 4.32% in order to study and obtain mechanical properties suitable for further uses. For comparison, three commercial alloys, VeraBond, Kera N, and VeraSoft, were used, with the properties taken from the specialized literature. They have been studied structurally and mechanically.

## 2. Materials and Methods

### 2.1. Materials Preparation

In this article, four alloys were used for testing and comparison. They were bought and are classic alloys that are currently used, while one alloy was modified to improve the properties: one Verabond alloy, one Kera N alloy, two VeraSoft alloys, and a VeraSoft alloy to which metallic silicon was added to.

The metal charge for making the alloy must be of high quality and purity, degreased, and properly prepared. A commercially purchased dental alloy of the VeraSoft type and high-purity metallic silicon, 99.6%, provided by Alfa Aesar from Thermo Fisher Scientific, were used as raw materials. In order to obtain the new VeraSoft alloy with a modified percentage, the use of an indirect heating furnace was chosen.

Two semi-finished products from the new alloy were obtained: a sufficient quantity for taking the samples needed for the laboratory tests that had in mind the proposed characterization. Figure 1 highlights the technological flow of developing the new VeraSoft alloy. The developed alloy was analyzed by structural and mechanical characterization methods.

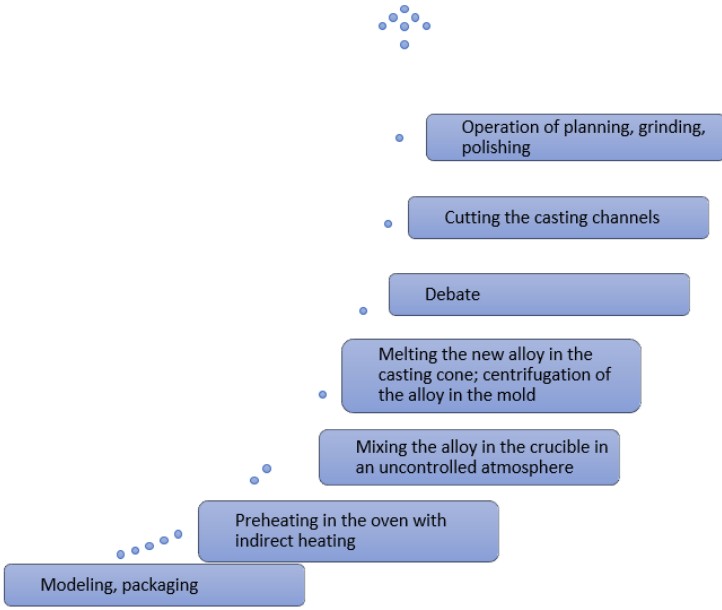

**Figure 1.** The technological flow of developing the new VeraSoft alloy.

In order to prepare the metallographic samples, a sequence of stages was completed as follows:

➢ The sanding process was performed with sanding papers. This was achieved by successive sanding operations on large-grained papers to small-grained papers.
➢ The polishing process with the help of felt. This was performed using aluminum oxide abrasive suspension of different grain sizes with the final goal of obtaining a mirror-gloss surface.
➢ The metallographic attack aimed to highlight the structural constituents. According to the ASM Handbook: 2 mL $FeCl_3$, 48 mL $H_2O$, and 50 mL HCL for 15 s at room temperature should be used.

VeraBond alloy (Figure 2a) is formulated for "Ceramco II" and other types of ceramics: Excelco, Spectrum, Noritake, Synspar, Shofu-Crystar, as well as with other medium-grained porcelains. The analyzed alloy has applications as a support for porcelain, and it is applied from a single element to full bridges, Maryland bridge, support for acrylate and composites from single elements to full bridge, all-metal crowns, and superstructures for implants.

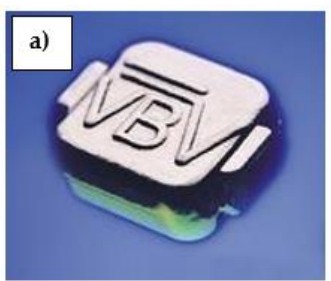 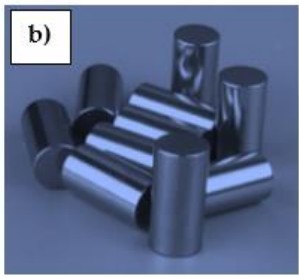 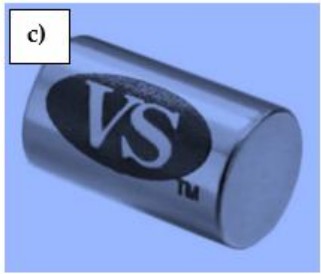

**Figure 2.** The commercial aspect of the analyzed samples: (**a**) VeraBond, (**b**) Kera N, (**c**) VeraSoft.

The KeraN alloy (Figure 2b) is an alloy of non-precious metals Nickel–Chromium, without Beryllium for ceramic work at high temperatures. It is suitable for open-fire processing and for dissociated working methods. An oxidizer is not required.

The VeraSoft (Figure 2c) alloy is a Ni-Cr casting alloy for metal crowns and bridges, pivots, restorations, and substructures for metal-acrylic crowns and bridges.

It is very important to recognize that synthetic materials have general and specific surface characteristics that depend on their properties. These characteristics must be known before any medical application, but they must also be known in relation to the changes that may occur over time in the body. In other words, any change in properties over time must be anticipated from the start and motivated by the choice of biomaterials and/or the device design.

The use of materials for tissue reconstruction in the human body dates back several thousand years, but clinically important advances have been achieved in the last century. The first interventions to replace some affected tissues date back more than 2000 years and are related to the Aztec, Chinese, and Roman civilizations [32,33].

### 2.2. Microstructural Characterization Methods

In order to obtain the most accurate determination, ten EDX determinations were performed in different areas of the semi-finished product to verify the concentrations obtained. To determine the chemical composition of the newly obtained alloy, the Vega Tescan LMH II equipment was used, using the Bruker EDAX detector attached to the SEM equipment.

The metallographic analysis provides information on the micrographic structure of the alloy, its nature, shape, dimensions, and distribution mode. A microscope Zeiss Axio Imager A1, was used for the optical analysis of high-precision optical images.

X-Ray diffractions (XRD, Panalytical, Almelo, The Netherlands) were conducted utilizing an Xpert PRO MPD 3060 facility from Panalytical (Almelo, The Netherlands) equipped with a Cu X-ray tube (K$\alpha$ = 1.54051), 2 theta: 10–70°, step size: 0.13°, time/step: 51 s, and a scan speed of 0.065651°/s in the reflection mode. Highscore Plus 3.0 software was used to analyze the data in order to determine the phase components and their parameters.

### 2.3. Mechanical Properties

The determination of the hardness of the commercial Ni-Cr alloys analyzed was carried out on a PMT3 model durometer by the Vickers method, using a pressing weight of 100 gf and a pyramidal diamond indenter.

Indentation is a test method based on the principles used to determine the modulus of elasticity, stiffness, etc. The indentation tests were performed using a device for tribological and mechanical determinations Universal Micro-Tribometer CETR UMT-2. For a more precise determination, three determinations were made for each individual alloy. After completing the working stages and recording them through the software of the UMT 2 device, the imprint curves (depth vs. force) of the new alloy obtained by the VIEWER program were plotted.

## 3. Results

### 3.1. Microstructural Analysis

Figure 3 shows the Spectrum EDX obtained and the new composition obtained after elaboration.

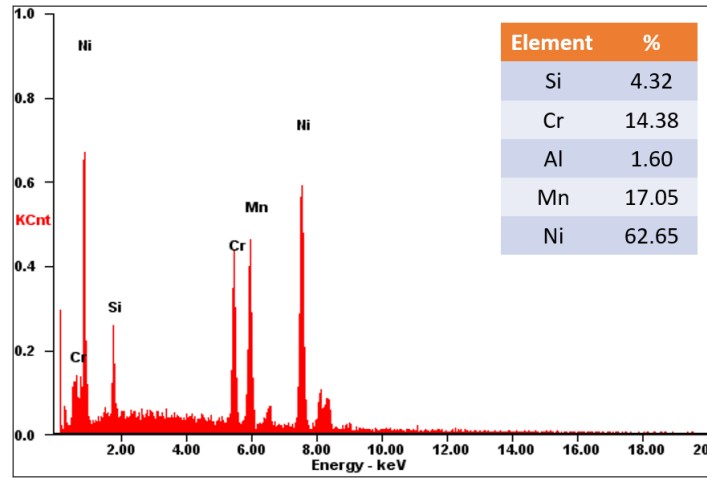

**Figure 3.** EDX spectrum for the newly obtained alloy with chemical composition obtained.

The chemical composition of the alloys is presented in Table 1 with average values for the three tests (±standard deviation) that were obtained regarding the chemical composition measurements results.

**Table 1.** The chemical composition of the alloys studied.

| | Ni [%] | Cr [%] | Mo [%] | Al [%] | Be [%] | Co [%] | Ti [%] | Si [%] | Mn [%] |
|---|---|---|---|---|---|---|---|---|---|
| VeraBond | 76.75 ± 0.20 | 12.60 ± 0.10 | 5.00 ± 0.10 | 2.90 ± 0.10 | 1.95 ± 0.10 | 0.45 ± 0.10 | 0.35 ± 0.10 | - | |
| Kera N | 61.60 ± 0.30 | 25.50 ± 0.10 | 11.00 ± 0.10 | 0.40 ± 0.10 | - | - | - | 1.50 ± 0.10 | |
| VeraSoft | 62.80 ± 0.10 | 14.50 ± 0.10 | - | 1.60 ± 0.10 | - | - | - | 1.60 ± 0.10 | 19.50 ± 0.10 |
| VeraSoft modified | 62.65 ± 0.20 | 14.38 ± 0.10 | - | 1.60 ± 0.10 | - | - | - | 4.32 ± 0.10 | 17.05 ± 0.10 |

The microstructure depends to a large extent on the alloying components but also on the processing conditions (melting, casting, and heat treatment).

Figure 4 presents the optical microstructure of the VeraBond, Kera N, VeraSoft, and the modified VeraSoft. Micrographs of the dental alloys are highlighted homogeneous dendritic structures.

VeraSoft and the modified VeraSoft have a dendritic structure, typical of cast alloys, which are solidified after medium to high-speed cooling. For the modified Verasoft (Figure 4d,h), the influence of the addition of the silicon element is very well observed; in this, it has refined the structure and contains, along with the dendritic phase, eutectic separations that are observable in the interdendritic space.

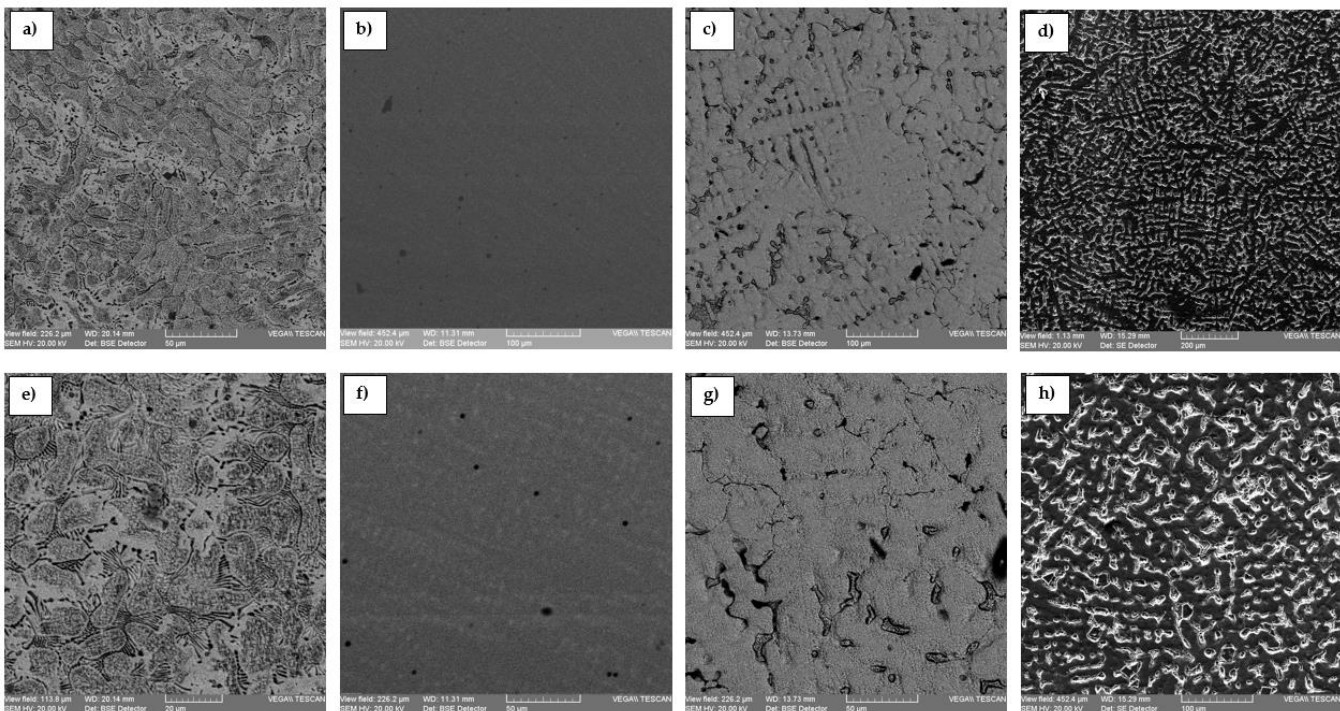

**Figure 4.** Microstructure analysis of experimental alloys: (**a**) VeraBond (200×), (**b**) Keran N (200×), (**c**) VeraSoft (200×), (**d**) VeraSoft modified (200×) (**e**) VeraBond (500×), (**f**) Kera N (500×), (**g**) VeraSoft (500×), (**h**) VeraSoft modified (500×).

The microstructure of dental alloys shows a homogeneous cellular structure with cellular boundaries. Similar findings were also reported by some previous reports [5,6]. This was due to the rapid solidification and strong temperature gradients of the melt during the melting process.

Usually, Ni-Cr alloys also contain, in smaller concentrations, Al, Co (hardening elements) and B, Si (deoxidizing elements). These alloys constitute a support for fused porcelain.

The alloys crystallize in the face-centered cube system, and the cast alloy has a dendritic structure with large grains.

The hardening of the alloy involves the precipitation of the subsequent phase or phases, particularly called the γ phase, consisting of $(NiCo)_3(AlTi)$. Carbs can form interdendritically. The values of the modulus of elasticity and hardness are somewhat lower than those presented by Co-Cr.

Solidification shrinkage is 1.5%, and the alloys are normally melted in induction furnaces and cast into phosphate forms. Due to the low-temperature range, Ni-Cr alloys offer a much more precise casting, which allows dental bridges and crowns to have minimal deviations [34,35].

### 3.2. Structural Analysis

XRD is a useful technique to study phases in dental alloys and their transformations as a function of temperature. Figure 5 presents the diffractograms of experimental alloys.

The phases revealed by XRD at the different analysis temperatures were in good agreement with those found in previous optical microstructures of transformations in these alloys. For the modified VeraSoft, new XRD peaks appear to result from the low-temperature transformation in martensite, which the optical microstructure has shown.

XRD diffractograms highlight that alloys exhibit three compounds: 2100373, Reference code: 96-210-0374, Chemical formula: Cr156.00 Al596.00; 9013031, Reference code: 96-901-3032, Chemical formula: Ni4.00; 1100049, Reference code: 96-110-0050, Chemical formula: Cr30.00Al48.00 [36].

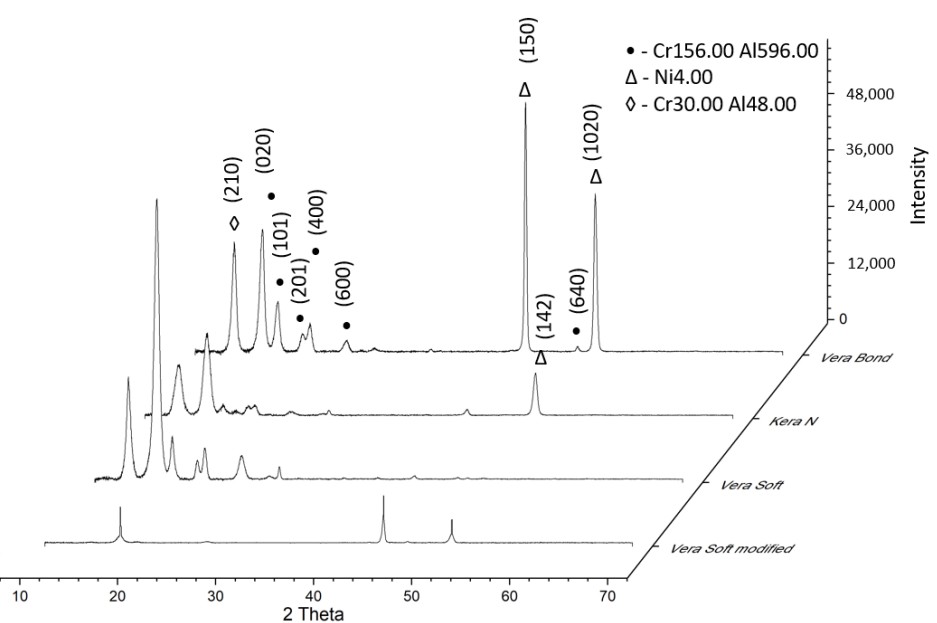

**Figure 5.** Diffractograms of experimental alloys.

These compounds that were formed in alloys were identified by XRD analysis. The main characteristics for Cr156.00 Al596.00 are as follows: crystal system: hexagonal, a (Å): 32.3000, b (Å): 32.3000, c (Å): 12.4000, alpha (○): 90.00, beta (○): 90.00, gamma (○): 90.00, calculated density (g/cm³): 3.59, and volume of the cell ($10^6$ pm³): 11203.59. The main characteristics for Ni4.00 are as follows: crystal system: cubic, a (Å): 3.5870, b (Å): 3.5870, c (Å): 3.5870, alpha (°): 90.00, beta (°): 90.00, gamma (°): 90.00, calculated density (g/cm³): 8.45, and volume of the cell ($10^6$ pm³): 46.15. The main characteristics for Cr30.00Al48.00 are as follows: hexagonal, a (Å): 12.7184, b (Å): 12.7184, c (Å): 7.9367, alpha (°): 90.0000, beta (°): 90.0000, gamma (°): 120.0000, calculated density (g/cm³): 4.26, and volume of the cell ($10^6$ pm³): 1111.8.

### 3.3. Hardness Results

For the dental alloys used in this study, Vickers hardness was determined by indentation tests. The samples were polished with 1 μm of alumina paste. After imprinting, the surface of the samples was processed, and the results were obtained tangentially to the surface with a Vickers indenter applied every 0.5 mm along the diameter of the sample. A load of 50 (Kg/mm²) and contact times of 15 s were used. On this basis, the average hardness, expressed in degrees Vickers (HV), was determined for each sample studied.

The Vickers hardness test involves a pyramid-shaped diamond indenter with a square base and a penetration angle of 136°; the load, in this case, is applied to the tested material constantly. The length of the penetrato's diagonals is used to determine the size of the indentation that it leaves in the material. The significant advantage of this test is that very small samples can be tested due to their small penetrator. Additionally, the ability of the test to work with varied tasks leads to another advantage; namely, hardness determinations can be made on both soft and hard materials. Vickers testing gives very good results in the case of brittle materials, but also, the situations in which ductile materials are encountered have been successfully solved.

Table 2 shows the average values for 10 tests (±Standard Deviation) obtained regarding the hardness measurement results of the dental alloys. The measurements showed values between 203 HV and 430 HV. As the percentage of silicon increases, the hardness decreases.

Modified VeraSoft has a lower hardness value compared to the classic VeraSoft alloy. The addition of silicon was beneficial and decreased the hardness. For dental applications, a very high hardness value is not necessary.

**Table 2.** The hardness values for alloys studied.

| Alloys | VeraBond | Kera N | VeraSoft | VeraSoft Modified |
|--------|----------|--------|----------|-------------------|
| HV | 429.5 ± 4.60 | 388.8 ± 5.10 | 251.5 ± 3.40 | 203.4 ± 5.70 |

### 3.4. Indentation Results

Metals and their alloys were the first materials used in human prosthetics, which, together with other inorganic and organic materials, are still widely used today. The use of metallic materials in one or another field of technology depends on the relationship between their structure and properties.

Micro indentation is a hardness testing technique for measuring the physical properties of samples, such as film layers that are often too small for conventional physical testing techniques.

The samples to be tested by micro indentation must be securely mounted and also have a relatively flat test surface. Samples are often prepared using the same techniques as the samples prepared for electron microscopy, requiring mounting in a hard matrix such as epoxy and then planar polishing of the test surface.

Figure 6 shows the response of the alloys during the indentation tests in the form of force-depth dependencies, and Table 3 presents the results of the indentation test with the average values for the three tests (± standard deviation) performed.

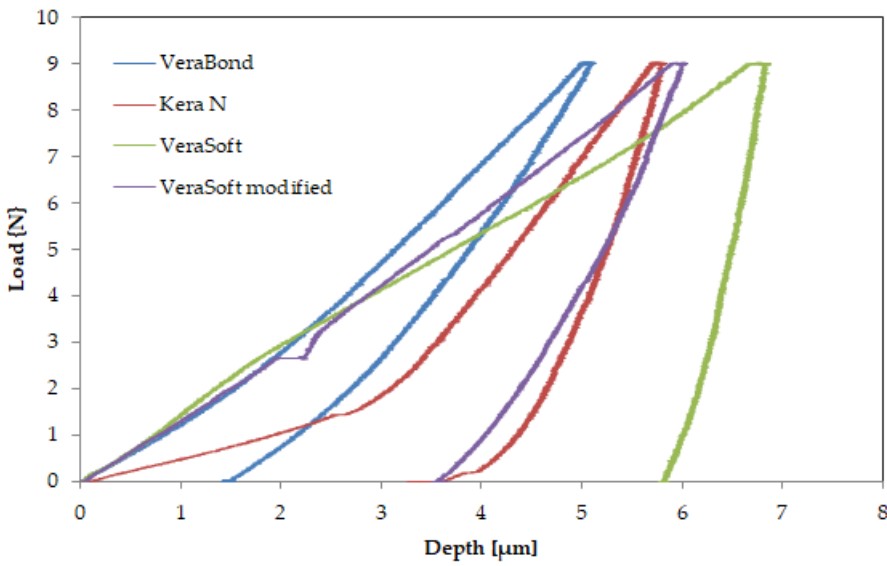

**Figure 6.** Indentation curves of the studied alloys: (a) VeraBond, (b) Kera N, (c) VeraSoft, (d) VeraSoft modified.

**Table 3.** Indentation results.

| Sample | Loading Deformation [N] | Release Deformation [μm] | Young Modulus [GPa] | Stiffness [N/μm] | Specimen Poisson Ration |
|--------|--------|--------|--------|--------|--------|
| VeraBond | 9.01 ± 0.20 | 5.12 ± 0.10 | 56.37 ± 0.50 | 3.314 ± 0.10 | 0.23 |
| Kera N | 9.01 ± 0.30 | 5.73 ± 0.20 | 152.95 ± 0.60 | 12.83 ± 0.30 | 0.23 |
| VeraSoft | 9.01 ± 0.10 | 6.84 ± 0.10 | 139.78 ± 0.90 | 13.05 ± 0.40 | 0.23 |
| VeraSoft modified | 9.02 ± 0.10 | 5.62 ± 0.30 | 46.79 ± 0.80 | 4.589 ± 0.10 | 0.23 |

The mean value ± standard deviation values of at least three independent specimens were used to obtain all the results. The authors [37] stated that Young's modulus could be predicted using Equations (1) and for the Poisson ration Equation (2).

$$E = \frac{(1 - v^2)}{2ak\left(\frac{a}{h},\ v\right)} \cdot \frac{P}{w} \tag{1}$$

$$\frac{P_1 w_2 a_2}{P_2 w_1 a_1} = \frac{k\left(\frac{a_1}{h}, v\right)}{k\left(\frac{a_2}{h}, v\right)} \tag{2}$$

Mechanical properties that characterize the behavior of metals and alloys under the action of external forces are the basis of their most important applications.

Because the mechanical properties of metals and alloys are structure dependent, they are very sensitive to the effects of manufacturing processes, and the same material can produce different properties.

The modulus of elasticity obtained by the indentation method for the dental alloys is in the range of 46–153 GPa. The lowest value is presented by the new-modified VeraSoft (46.79 GPa). The addition of silicon had a beneficial effect on the mechanical properties, reducing the modulus of elasticity

The biomaterials used in oral implantology must ensure the transmission of occlusal forces to the supporting tissues. For this reason, they must present sufficient mechanical resistance but also an elasticity adapted to the bone. The functional biocompatibility of the implant material mainly concerns the modulus of elasticity and mechanical strength. The elasticity mode is the characteristic with significant influence on the bone remodeling process, causing the destruction of the tissue-implant assembly in the conditions of a major difference between the components of the implant-bone assembly [38–40].

A difference between the elastic modulus causes a mechanical stress on the tissue that is different from the physiologically normal one. The finite element analysis of the behavior of the bone-implant assembly under mechanical stress indicated that a material with a small modulus of elasticity could determine a distribution much closer to that of the mechanical stresses in the surrounding bone tissue.

The newly developed alloy aims to be used in medical applications, with the role of fulfilling the functional requirements and removing some disadvantages in the classic alloys used as biomaterials in human tissue.

The superior valorization of metallic biomaterials requires knowledge of the chemical, physical, mechanical, thermal, electrical, magnetic, optical, as well as technological properties by means of the specific methods of these categories of materials.

## 4. Conclusions

The performance of the field and the condition of success in oral implantology require today the review and reevaluation of the means that contribute essentially to ensure the stability and durability of the implant, starting from the nature of the biomaterial and continuing with the characteristics of the optimally designed biosurface.

Dental metal alloys have advantages such as a pleasant appearance, low price (the non-precious ones), and good physical and mechanical properties (such as high modulus of elasticity), which allow the use of smaller sections of the alloy, and therefore, less destruction to the teeth during crown preparation, They also provide an adequate coefficient of thermal expansion comparable to that of porcelain commonly used for veneering, which maintains the metal and crown ceramics which are intimately bonded during casting and prevents the cracking of the coating.

Precious and semi-precious dental alloys are generally recognized as inert and perfectly biocompatible. Some metals are already considered prohibited for organisms, such as beryllium, nickel, and mercury.

All components of the prosthetic field undergo changes over time, and the reactions of the oral mucosa are the result of mechanical irritation, plaque accumulation, as well as the toxic and/or allergic action of the materials from which the implants are made. For this reason, we must carefully choose the best materials so that once introduced into the human body they do not affect its health.

The aim of this paper consists of a comparative analysis of three commercial alloys, VeraBond, Kera N, and VeraSoft, compared to a modified dental alloy, with the aim of improving some mechanical properties.

Regarding the microstructural analyses, alloys show that the alloys crystallize in the face-centered cube system, and the cast alloy has a dendritic structure with large grains. XRD diffractograms highlight that alloys exhibit three compounds Cr156.00 Al596.00: 9013031, Ni4.00, and Cr30.00Al48.00.

For the mechanical properties, hardness measurements show values between 203 HV and 430 HV. As the percentage of silicon increases, the hardness decreases. The results of the indentation test for the dental alloys were in the range of 46–153 GPa. The results show that the hardness and elastic modulus of the new alloy was significantly minimized compared to the classical alloys used.

The percentage of silicon increased the hardness value and decreased the modulus of elasticity in the case of the new alloy obtained. The newly obtained alloy lends itself to uses in the medical field; the examinations performed to reveal a suitable behavior for prosthetic constructions.

**Author Contributions:** D.C.A.: conceptualization, management; M.S.B.: writing—original draft preparation, investigation; P.V.: methodology, financing, data curation; A.V.S.: investigation, writing—review and editing; M.B.: validation, investigation; B.I.: investigation, data curation. All authors have read and agreed to the published version of the manuscript.

**Funding:** This paper was financially supported by the Project "Network of excellence in applied research and innovation for doctoral and postdoctoral programs/InoHubDoc", project co-funded by the European Social Fund financing agreement no. POCU/993/6/13/153437. This paper was also supported by "Gheorghe Asachi" Technical University from Iaşi (TUIASI), through the Project "Performance and excellence in postdoctoral research 2022".

**Institutional Review Board Statement:** Not applicable.

**Informed Consent Statement:** Not applicable.

**Data Availability Statement:** All data provided in the present manuscript are available to whom it may concern.

**Conflicts of Interest:** The authors declare no conflict of interest.

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
