# Peer review of "Ni-Cr Alloys Assessment for Dental Implants Suitability"

_applsci, doi:10.3390/app122412814_

Round 1
Reviewer 1 Report
The paper of Achitei and co-worker entitled ''Ni-Cr alloys assessment for dental implants suitability'' as a whole contains interesting data but needs to be improved after taking into account the comments below.
The article requires major revision.
[1] Authors must provide information regarding medical materials applications such as biocompatibility and cell toxicity.
[2] The mechanical properties you are presenting do not show any std I also think that fracture toughness should be more evaluated as that was the reason this work was made and for that.
[3] Add more XRD experimental details including the previous citations and determined whether the XRD patterns are reflection or transmission mod?
[4] The materials & methods part needs to be improved, all used materials need to be described. Also, cite the work you have followed.
[5] Authors better replace software-generated SEM scale bars with smaller scale bars to show more detail.
Author Response
Dear reviewer,
We are grateful for the time you have taken to evaluate our manuscript. We are thankful for your comments. They helped us to improve the quality of our paper. Responses to specific issues are given below. The changes in the manuscript have been highlighted by yellow. We hope the paper has been improved and is now acceptable for publication in Applied Science.
Kind regards,
Authors

Reviewer 2 Report
The manuscript describes a new method to improve a commercial metalic alloy for dental implants. Some experimental details are missing in methodology section. The results are little discussed.
My suggestions are:
1. Line 140/141: add information such as concentration, temperature, time
2. Figure 2 can be moved to results section together with Table 1.
3. Add equation to obtain Young modulus and Poisson ratio
4. Line 349/350, the authors claim "The results showed that the hardness, elastic modulus of the new alloy was significantly increased compared to classical alloys used" However, the results showed that the modified Verasoft has a lower hardness value. Please verify.
Author Response

(The authors gave the same response as above.)

Reviewer 3 Report
Dear Authors,
Please find the attached.
Best Regards

Author Response

(The authors gave the same response as above.)

Round 2
Reviewer 1 Report
The corrections that were asked for have been made and the manuscript is acceptable in this form.
Reviewer 2 Report
The authors took into account my suggestions and improved the manuscript.